# Early Response Assessment to Targeted Therapy Using 3′-deoxy-3′[(18)F]-Fluorothymidine (^18^F-FLT) PET/CT in Lung Cancer

**DOI:** 10.3390/diagnostics10010026

**Published:** 2020-01-06

**Authors:** Kalevi Kairemo, Elmer B. Santos, Homer A. Macapinlac, Vivek Subbiah

**Affiliations:** 1Department of Nuclear Medicine, The University of Texas M.D. Anderson Cancer Center, Houston, TX 77030, USA; Kalevi.Kairemo@gmail.com (K.K.); hmacapinlac@mdanderson.org (H.A.M.); 2Investigational Cancer Therapeutics, The University of Texas M.D. Anderson Cancer Center, Houston, TX 77030, USA; vsubbiah@mdanderson.org

**Keywords:** ^18^F-FLT, ^18^F-FLT PET/CT, lung cancer, response to therapy, small cell lung cancer, non-small cell lung cancer

## Abstract

Although 2-deoxy-2-[18F]-fluoro-D-glucose positron emission tomography/computed tomography (^18^F-FDG PET/CT) is a sensitive nuclear medicine modality, specificity for characterizing lung cancer is limited. Tumor proliferation and early response to molecularly targeted therapy could be visualized using 3′-deoxy-3′[(18)F]-fluorothymidine (^18^F-FLT) PET/CT. The superiority of ^18^F-FLT PET/CT over ^18^F-FDG PET/CT in early therapeutic monitoring has been well described in patients after chemotherapy, radiotherapy, and/or chemo/radiotherapy. In thispilot study, we explorethe use of ^18^F-FLT PET/CT as an early response evaluation modality in patients with lung cancerand provide specific case studies of patients with small cell lung cancer and non-small cell lung cancer who received novel targeted therapies. Early response for c-MET inhibitor was observed in four weeks and for MDM2 inhibitor in nine days.

## 1. Introduction

^18^F-fluorothymidine, [^18^F]3-deoxy-3-fluorothymidine (^18^F-FLT), is an imaging biomarker of cellular proliferation and has been utilized in various cancer types including blood (lymphoma and leukemia), breast, head and neck, esophageal, and lung cancers as well as in bone and soft tissue sarcomas [1,2,3,4,5]. ^18^F-FLT is derived from the cytostatic drug azidovudine developed for positron emission tomography (PET) imaging [6]. Its main role is in evaluating treatment response [1] by indirect monitoring of cell proliferation based on DNA synthesis. ^18^F-FLT is basically a radiolabeled structural analog of thymidine, which is a constituent nucleoside of DNA. Although ^18^F-FLT is not incorporated into DNA (or at least selectively), it reflects the level of DNA synthesis because of its entrapment inside the cell through phosphorylation by thymidine kinase-1 (TK1) expressed during the S-phase of DNA synthesis [7]. Precisely, ^18^F-FLT uptake in malignant cells correlate with activity of TK1 which is usually low in the cell resting stage but high in the deregulated cancer cell cycle.

In contrast with the most widely-used radiotracer for PET/CT imaging, the 2-deoxy-2-[18F]-fluoro-D-glucose (^8^F-FDG), ^18^F-FLT shows a lower accumulation in tumors than ^18^F-FDG as it only accumulates in cells that are in the S-phase of growth and demonstrates a low sensitivity for nodal staging in lung cancer. In addition, only 8–20% of the cells are in the S-phase, so in comparison to glucose metabolism and ^18^F-FDG accumulation, absolute accumulation of TK1-catalyzed radioactivity is relatively low. In spite of these limitations, ^18^F-FLT has shown to be superior in imaging proliferation [8,9,10]. In a meta-analysis of 27 articles involving 1213 lung cancer patients for correlating FDG uptake (22 studies) or FLT uptake (eight studies) with Ki-67 expression, the rho coefficient for ^18^F-FDG/Ki-67 and ^18^F-FLT/Ki-67 was 0.45 (95% CI, 0.41–0.50) and 0.65 (95% CI, 0.56–0.73), respectively, which indicated a moderate correlation for ^18^F-FDG and a significant one for ^18^F-FLT [8]. 

It has been shown that shortcomings of ^18^F-FLT and ^18^F-FDG can be theoretically solved with dual tracer imaging studies [11,12,13,14,15]. For instance, in 55 patients with pulmonary nodules who underwent ^18^F-FDG PET/CT and ^18^F-FLT PET/CT within seven days, the sensitivity and specificity for ^18^F-FDG PET/CT was 87.5% and 58.9% and for ^18^F-FLT PET/CT was 68.7% and 76.9%, respectively. The combination of ^18^F-FLT and ^18^F-FDG improved sensitivity up to 100% and specificity up to 89.7% [13]. This dual tracer imaging induced substantial change in clinical management of 31.5% of patients with pulmonary lesions and partial change in another 12.3% [14,15]. In a meta-analysis pooled from 17 studies [12] including 548 patients with malignant and benign lung lesions (bronchioloalveolar lung carcinoma, squamous cell carcinoma, non-small cell lung cancer, small cell lung cancer, adenocarcinoma, tuberculosis, fibromas, hamartomas, etc.), it showed that although ^18^F-FLT cannot replace ^18^F-FDG in detecting small cell lung cancer and early development of lung cancer, it may help to prevent patients with misdiagnosis of inflammatory lesions.

To date, there are no clinical studies of ^18^F-FLT PET/CT of novel targeted therapiesin assessing early response in lung cancer. c-MET inhibitors, have the potential to benefit subsets of lung cancer patients with specific genetic alterations [16]. Exon-14 skipping mutations appear so far to be the most promising molecular subset that is sensitive to c-MET inhibitors, whereas overexpression, amplification, and point mutations of MET seem more challenging subgroups to target [17]. Combination with other target agents, such as EGFR inhibitors, may represent a promising therapeutic strategy in specific areas (e.g., EGFR-TKI resistance), because HGF/c-MET pathway mediates VEGFR inhibitor resistance and vascular remodeling in NSCLC [18]. Mouse double minute 2 protein (MDM2) is a regulator of tumor suppressor P53. Inhibitors of MDM2 are in clinical development. 

In this pilot study, we evaluated the early treatment response for. c-MET inhibitor and mdm-2 inhibitor in lung cancer. We show that early assessment of therapy response from these two new drugs was feasible with ^18^F-FLT PET/CT imaging and that the combination with ^18^F-FDG PET/CT would have more potential.

## 2. Materials and Methods

We reviewed the medical records of patients with advanced lung cancer (small cell and non-small cell lung cacer) who had FLT/PET imaging as part of their care at MD Anderson. This study was performed in accordance with the guidelines of the MD Anderson Institutional Review Board (IRB). Because this was a retrospective chart review IRB has waived the consent requirements. They were enrolled on c-MET inhibitor and MDM2 inhibitor based trials available in the institution. This study was compliant with the Health Insurance Portability and Accountability Act. Written informed consent was obtained from each participant for enrollment on the respective clinical trials.

### 2.1. PET/CT Study

PET/CT studies were performed using a Discovery ST8 PET/CT system (GE Healthcare) in combination with the CT, 8-MDCT scanner (LightSpeed, GE Healthcare). ^18^F-FLT (200–350 MBq) or ^18^F-FDG (10–15 MBq) was administered intravenously in a single dose injection. Whole-body PET imaging (WB PET) consisted of four or five 10 min bed positions, approximately 60 min from injection. PET images were reconstructed using standard vendor-provided algorithms. The CT consisted of a helical scan covering the head to the mid thighs (120 kVp, 300 mA, 0.5-s rotation; table speed, 13.5 mm/rotation) with no contrast enhancement. Axial CT slice thickness was 3.75 mm. The PET data was corrected for random coincidences, scatter, and attenuation. Transaxial images were reconstructed into 128 × 128 pixel matrices with a pixel size of at least 4.5 mm, using vendor-provided algorithms that incorporate ordered-subset expectation maximization and were corrected for attenuation using CT data; the emission data was corrected for scatter, random events, and dead-time losses as well, using the PET/CT scanner’s standard algorithms. Measures of uptake and retention in tumors were obtained from the WB PET data and compared to normal tissue.

### 2.2. ^18^F-FLT PET/CT Scan Interpretation

Blinded image interpretation was performed by two experienced nuclear medicine physicians (KK, EBS) with more than 20 years of experience in the field. ^18^F-FLT PET/CT uptake of target lesions was evaluated visually as present or absent, graded on a 3-point scale. After the final interpretation/score entered for each patient, the same reviewers determined the maximum standardized uptake value (SUV) for each residual mass/lesion by CT, regardless whether it was FLT-PET negative. 

## 3. Results

Presented here are case studies of patients with biopsy-proven lung cancer in a clinical trial for ^18^F-FLT PET/CT monitoring of treatment response. Table 1 summarizes the results of these studies.

Patient 1: The first patient in the case study is a 49-year-old male with small cell lung cancer who received cisplatin and etoposide as standard therapy. After progression from standard therapy, the patient was placed on a c-MET inhibitor therapy. We compared conventional CT, FLT-PET/CT, and FDG-PET/CT scans pre- and post-four weeks of targeted therapy (Figure 1). In this patient, the pre-therapy tumor CT measurements were 1.5 cm × 0.7 cm, which remained unchanged post therapy after four weeks. While the ^18^F-FDG PET/CT showed a 2% increase in SUVmax from 1.64 to 1.67, ^18^F-FLT PET/CT showed a reduction in metabolic activity of 38% from 1.3 to 0.8 SUVmax (Figure 1).

Patient 2: This is a 51-year-old female with a history of relapsed and refractory lung adenocarcinoma. This patient received several lines of therapy that included erlotinib, carboplatin, gemcitabine/docetaxel, pemetrexed, sorafenib, irinotecan, and then bevacizumab. After progression on standard therapy, the patient was enrolled in a c-MET inhibitor-targeted-therapy clinical trial. At that time, she presented with a right hilar mass (Figure 2). The two-dimensional CT tumor measurements were 5.2 cm × 4.6 cm and post therapy were 5.3 cm × 4.8 cm. The change corresponds to a 2% increase in tumor size. However, the ^18^F-FLT PET/CT imaging pre- and post-therapy after four weeks revealed a SUVmax of 5.1 and 3.9, respectively, which was a 24% reduction in ^18^F-FLT activity. Although not significant statistically, it was a clinically significant response to therapy. 

Patient 3: The third case study is a 59-year-old female with a diagnosis of EGFR-negative lung adenocarcinoma. The patient received cisplatin, vinorelbine, pemetrexed, and cetuximab as part of standard therapy and was eventually enrolled on an MDM2 inhibitor. Herein, a nine-day pre- and post-targeted therapy image of the right middle lobe mass comparison was done using CT and ^18^F-FLT PET/CT. While CT demonstrated a change of 6% tumor size from 5.1 cm × 4.4 cm to 4.8 cm × 4.6 cm, ^18^F-FLT PET/CT showed a reduction of 31% from 3.6 to 2.5 SUVmax (Figure 3). 

## 4. Discussion

^18^F-FLT PET/CT in monitoring treatment response in patients was evaluated in patients with lung cancer. We demonstrate early/signals of activity with c-MET in four-weeks post therapy and another case with MDM2 inhibitor in nine-days post therapy. The favorable change in ^18^F-FLT activity was not evident in ^18^F-FDG PET/CT and CT.

The advantages of ^18^F-FLT PET/CT over ^18^F-FDG PET/CT in monitoring treatment response have been well described in patients after chemotherapy, radiotherapy, and/or chemo/radiotherapy. Allen et al. showed significant decrease in tracer uptake in non-small cell lung cancer and mesothelioma patients at the end of treatment with cisplatin and pemetrexed suggesting that ^18^F-FLT is a more specific tracer for changes in proliferation than ^18^F-FDG [19]. This theory is supported by known pharmacology of arginine depletion in argininosuccinate synthetase 1 -deficient tumor [20]. In a head-to-head comparison of responders vs. non-responders, patients with advanced adenocarcinoma of the lung on gefitinib, an EGFR tyrosine kinase inhibitor, showed significant decrease in tracer uptake seven days after the start of therapy [19]. Maximum tumor standardized uptake value (SUV max) declined by a mean of 36% in responders, compared to a 10.1% increase in non-responders. ^18^F-FLT PET/CT responders had a median time to progression of 7.9 months in comparison to 1.2 months in non-responders.

^18^F-FLT can also be a suitable tracer for radiotherapy monitoring. Uptake of ^18^F-FLT can be significantly reduced after 5–11 treatment fractions [21]. More recently, different studies have shown that ^18^F-FLT is a more sensitive tracer of early chemo/radiotherapy treatment response over ^18^F-FDG [22,23,24]. Unfortunately, false positive findings of ^18^F-FLT PET/CT may also occur. The reason is ^18^F-FLT uptake by proliferating lymphoid cells in germinal centers of reactive lymph nodes [25,26]. However, further studies are needed to fully evaluate the role of FLT and FDG. For now, we would not advocate the use of FLT for tumor detection or staging. Early response evaluation by FLT/PET remain exploratory as described and will need CT confirmation.

## 5. Conclusions

^18^F-FLT PET/CT for early assessment of therapy response to evaluate for signals of preliminary activity of novel targeted therapy in patients with lung cancer is feasible. Future studies are warranted to compare FLT-PET with conventional modalities. 

## Figures and Tables

**Figure 1 diagnostics-10-00026-f001:**
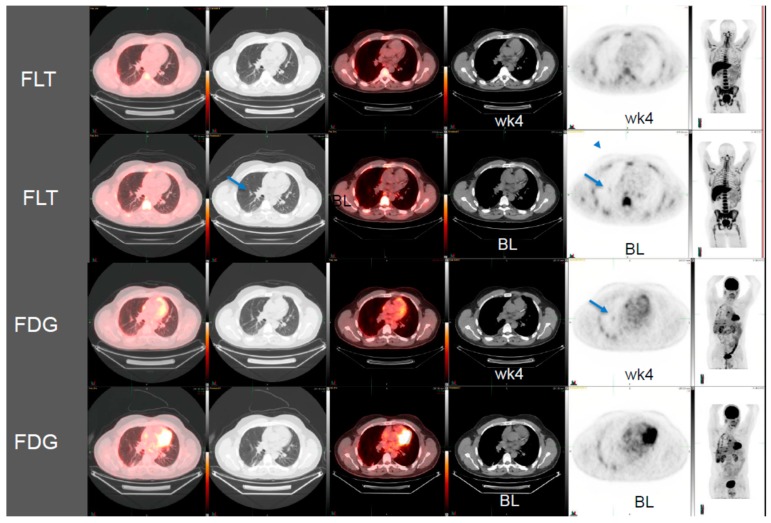
3′-deoxy-3′[(18)F]-fluorothymidine (^18^F-FLT) positron emission tomography/computed tomography (PET/CT) scans shows early response to c-MET inhibitor targeted therapy whereas the conventional CT scan and 2-deoxy-2-[18F]-fluoro-D-glucose (^18^F-FDG) PET/CT images do not show any responses in this patient with small cell lung cancer (BL, baseline; wk4, at 4 weeks post therapy).Arrows show the location of tumors.

**Figure 2 diagnostics-10-00026-f002:**
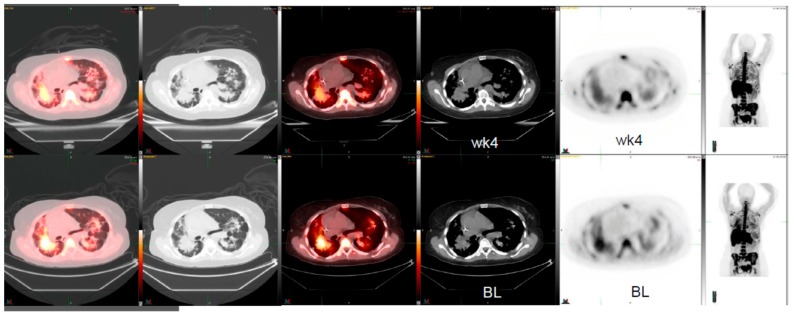
^18^F-FLT PET/CT scans shows weak signals of early response to MET inhibitor targeted therapy whereas the conventional CT scan images do not show any responses in a patient with lung adenocarcinoma (BL, baseline; wk4, at 4 weeks post therapy).

**Figure 3 diagnostics-10-00026-f003:**
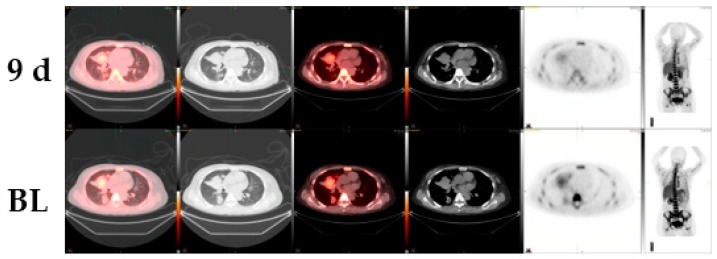
^18^F-FLT PET/CT scans shows early response to MDM2-inhibitor targeted therapy whereas the conventional CT scan images do not show any responses in a patient with EGFR-negative lung adenocarcinoma (BL, baseline; 9d, at 9 days post therapy).

**Table 1 diagnostics-10-00026-t001:** Summary of the study. Patient characteristics (age, gender, diagnosis), new therapy, previous therapies, and the scan changes in the current study (FLT-PET/CT, FDG-PET/CT and CT).

Age/Gender/ New Therapy	Diagnosis	Previous Chemotherapy	^18^F-FLT SUV Change	Comments
49/male c-MET-inhibitor	small cell lung cancer	cisplatin, etoposide	−38%	FDG-change +2% CT-change 0 %
51/female c-MET-inhibitor	lung adenocarcinoma	erlotinib, carboplatin, gemcitabine/docetaxel, pemetrexed, sorafenib, irinotecan, bevacizumab	−24%	EGFR positive CT-change +2%
59/female MDM2 inhibitor	lung adenocarcinoma	cisplatin, vinorelbine, pemetrexed, cetuximab	−31%	EGFR negative CT-change -6%

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
