# Peer review of "Early Response Assessment to Targeted Therapy Using 3′-deoxy-3′[(18)F]-Fluorothymidine (18F-FLT) PET/CT in Lung Cancer"

_diagnostics, 2020, doi:10.3390/diagnostics10010026_

Round 1

Reviewer 1 Report

The manuscript titled," Early Response Assessment to Two Targeted Therapies by 18F-FLT PET/CT in Lung Cancer", authored by Kairemo et al., is addressing clinically relevant and significant topic. Performing dual tracer imaging using FDG-PET and FLT-PET scans and matching the both to get some conclusive findings is an important and relatively novel strategy. The manuscript is scientifically sound and can be published in "Daignostics" journal. However, authors must pay attention to the language (grammar, formatting and uniformity) as well as legends for images which are not clear. What BL stands for? Figure 3 requires proper labeling. Introduction had sufficient background information, however, authors just mentioned about this work in single sentence, which can be elaborated.

Author Response

We graciously thank the reviewers for their time and expertise in giving us constructive critiques to potentially publish this paper in Diagnostics. We agree in their comments and hope that our replies   and amendments be at least satisfactory for publication of our studies. 

Our replies are itemized below according to the comments made by the reviewers.

Authors must pay attention to the language (grammar, formatting and uniformity) as well as legends for images which are not clear. What BL stands for? Figure 3 requires proper labeling.

The writing has been improved including grammar correction and proper format application. In addition, scientific terms, chemical symbols, and radiopharmaceutical nomenclatures were now uniformly written. Typographical errors were corrected and proper figure legends were placed. 

Introduction had sufficient background information, however, authors just mentioned about this work in single sentence, which can be elaborated.

We agree with the reviewer’s comment about more information of this study. Further elaboration of this work was done and the background information was made more succinct. 

Reviewer 2 Report

The overall idea is interesting but a few pointers:

1) The description of the patients as in patient#1, 2,3 etc. with their subsequent treatment should be also presented in a table format so its easy to refer.

2) Also if the discussion could be more elaborate on the importance of using this particular two targeted therapy

Author Response

We graciously thank the reviewers for their time and expertise in giving us constructive critiques to potentially publish this paper in Diagnostics. We agree in their comments and hope that our replies   and amendments be at least satisfactory for publication of our studies. 

Our replies are itemized below according to the comments made by the reviewers.

1) The description of the patients as in patient#1, 2,3 etc. with their subsequent treatment should be also presented in a table format so its easy to refer.

We agree with the reviewer’s comment to summarize patient’s profile for clarity of presentation. We have added a table containing patients’ age, gender, diagnosis, new therapy, previous therapy, change in FLT SUV, and comments.

2) Also if the discussion could be more elaborate on the importance of using this particular two targeted therapy.

Further elucidation of this work has been added. The importance of using these two targeted therapies in this project has been elucidated and added in the discussion.